# Effects of Changing Veterinary Handling Techniques on Canine Behaviour and Physiology Part 1: Physiological Measurements

**DOI:** 10.3390/ani13071253

**Published:** 2023-04-04

**Authors:** Camille Squair, Kathryn Proudfoot, William Montelpare, Karen L. Overall

**Affiliations:** 1Department of Health Management, Atlantic Veterinary College, University of Prince Edward Island, Charlottetown, PE C1A 4P3, Canada; 2Department of Applied Human Sciences, Faculty of Nursing, University of Prince Edward Island, Charlottetown, PE C1A 4P3, Canada

**Keywords:** low-stress handling, veterinary examination, collaborative care, animal welfare

## Abstract

**Simple Summary:**

Canine patient stress during veterinary visits is an animal welfare and health concern. Low-stress handling, combined with cooperative and collaborative care, has been proposed to reduce patient fear; however, research investigating these techniques in a veterinary setting is lacking. The aim of this study was to determine whether targeted interventions during veterinary visits helped to prevent or reduce distress in dogs. A total of 28 dogs were examined within four visits across 8 weeks. Following the first visit, dogs were split into intervention and control groups, where the intervention group received simple low-stress handling techniques and practiced collaborative care, and the control groups experienced routine care. The interventions were selected so that they could easily be incorporated into a busy veterinary hospital. There was a greater reduction in serum cortisol, an indicator of stress, between the first and last visit in the intervention group. The intervention group had a significant reduction in a composite stress response index from the first to last visit. Our findings have important applications both for dogs who are already afraid of veterinary examination and for use in a preventative context with dogs just beginning their veterinary experience.

**Abstract:**

Signs of distress in dogs are often normalized during routine veterinary care, creating an animal welfare concern. We sought to test whether targeted interventions during veterinary visits affect physiological indicators of stress in dogs. Some 28 dogs were examined within four visits across 8 weeks. All dogs received the same care during the first visit and were then randomized into control and intervention groups for visits 2–4. In the intervention group, 14 dogs underwent procedures designed to reduce stress and to enlist their collaboration during examination. The 14 dogs in the control group received routine care. At each visit, heart rate (HR), serum cortisol (CORT), neutrophil lymphocyte ratio (NLR), and creatine kinase (CK) were measured. A composite stress index based on the summed standardized scores for these markers was constructed. No differences in HR, NLR, and CK parameters between groups were found, and both groups had a decrease in CORT by visit four. However, the intervention group showed a greater overall decrease in CORT between the first and fourth visit than the control group (*p* < 0.04). The composite stress index differed between the first and fourth visits for the intervention group, but not for the control group (Intervention *p* = 0.03; Control *p*= 0.288). There was a tendency for the composite stress index to worsen at visit four vs. visit one for the control group. The findings suggest that dogs that participated in adaptive, collaborative exams and procedures designed to minimize fear had a greater reduction in stress over time compared to those receiving standard care.

## 1. Introduction

The welfare of dogs undergoing veterinary care has primarily been focused on achieving a basic standard of physical care that includes redressing pain. Through this lens, signs of distress have often been normalized—and in many cases, expected—as part of routine care, when instead, they should be viewed as welfare concerns. Fear has been demonstrated to begin as soon as dogs walk into a clinic. For example, a study conducted at a veterinary hospital in Germany found that fewer than half of the dogs entered the clinic calmly, and 13.3% had to be dragged or carried into the building [1]. A similar finding was highlighted in a study demonstrating that when walking into a vet clinic, 60% of dogs showed apprehensive postures and 18% showed signs of fear-related aggression [2]. Two-thirds of dogs in a veterinary waiting room spent more than 20% of the time exhibiting at least one sign of stress, and 53% exhibited four or more signs of stress [3]. Other aspects of veterinary visits including walking onto a scale and physical examination have been shown to increase stress in canine patients [1,4,5,6]. Multiple studies have shown that a majority of dogs show signs of fear while on an examination table [1,5], and a recent study found that fear responses were increased during examination that involved physical manipulations [7]. The pervasive level of fear and anxiety currently experienced by patients during veterinary visits has prompted immediate welfare concerns in addition to long term implications and consequences. Each negative event an animal experiences when at the veterinary clinic conditions them for the future negative responses to similar events, causing subsequent visits to become more difficult and time-consuming [1,8]. Dogs < 2 years old that visited the practice frequently were more fearful than older dogs that visited infrequently [1], suggesting that exposure to experiences they perceive as fearful matters to them, and that repeated exposure to veterinary practices may actually enhance fear.

Fear and anxiety can have significant physiological consequences for dogs. In many species, including dogs, distress can reduce immune function and reproductive abilities, increase the risk of contracting infectious diseases, delay healing, and have a negative effect on life span [9,10,11,12,13]. Exposure to stressors can further negatively affect treatment outcomes by causing pet guardians to delay intervention and preventative veterinary care. The stress experienced by the pet—and owner—and induced by veterinary appointments is a large contributor to decreased veterinary care [14]. A quantitative survey of 2188 dogs and cat owners found that 58% of cat owners and 38% of dog owners perceived that their animal “hates” going to the veterinarian, and 38% of cat owners and 26% of dog owners found it stressful just thinking about taking their animal to the veterinary clinic [14]. Stress responses experienced during veterinary visits affect the animal, pet guardian and veterinary staff in a variety of manifestations. These responses may include how frequently owners seek veterinary care, the negative effect of stress on the patient’s long-term health, reductions in the veterinarian’s ability to assess and accurately diagnose and treat health concerns, and the reduced safety of the veterinary staff, with an increased risk of injury associated with stressed animals [8,10,14,15].

Recently, there has been increased acknowledgment and awareness of the persistence of stress and fear in veterinary patients. With this acknowledgement, in the veterinary community, recognition of the importance of low-stress handling in the veterinary hospital and clinic has grown [15,16,17,18,19]. Some limited research has shown that dogs with positive experiences while at the veterinary clinic have been found to be less fearful than others [1]. Additionally, recent research has begun to investigate the influence of collaborative care on dog fear levels during veterinary examination [20]. Early veterinary visits set the foundation for subsequent interactions, and can have lasting effects on anxiety levels in patients [15,21]. However, data assessing the effect of the implementation of low-stress techniques within a veterinary setting and their effectiveness in reducing distress in patients are lacking. The aim of our study was to determine whether simple interventions, which could be easily implemented in any veterinary practice, affected measurements of distress at the veterinary clinic. Rather than singling out one intervention to assess, we used an examination protocol that altered most of the patterns of the standard physical exam for the intervention group, since what the dogs experience is the entire approach. Accordingly, we assayed responses to an overall pattern of changes, rather than to any of our individual interventions. 

This paper is part of a larger project in which we sought to evaluate all dogs behaviourally and physiologically for their responses to the veterinary visit. To measure the physiological response to the stress of the veterinary visits, we sought an approach that would evaluate various axes of the dogs’ responses to the stressor. We chose four parameters to evaluate: heart rate (HR), serum cortisol (CORT), serum creatine (phospho)-kinase (CK/CPK), and neutrophil/lymphocyte ratio (NLR). 

HR is a marker of an immediate sympathetic response [22], cortisol is a marker of an acute stress response [23], and NLR is a reliable immunological measure of chronic stress [24]. NLR is relatively unaffected by handling associated with acute stress such as blood sampling [24], and may be a good marker of sub-clinical inflammation [25]. CK can be an indicator of muscle tension and/or damage. In profound panic, acute muscle contraction and injury may be one component of the arousal and stress response associated with freezing [26], and so CK was selected as a component of a multi-modal assessment of physiological stress to evaluate any panic component [27]. 

## 2. Materials and Methods

This study was conducted from June to September 2021. Data were analyzed using SAS, Social Science Statistics (https://www.socscistatistics.com/ (accessed on 27 December 2022)), R and Excel (R Project, 2011; R version 4.1.2 (2021-11-01)—“Bird Hippie”). This research was approved by the Animal Care Committee (ACC) and the Research Ethics Board (REB) of UPEI (Joint Protocol 21-02). All dog guardians gave informed consent and could withdraw at any time.

### 2.1. Enrolment

A total of 30 dogs were enrolled for the study design target of 28 participants (Table 1) based on initial power calculations of 80% with a one-tailed probability of 0.1 and z_beta_ = 1.28. Dogs whose guardians expressed an interest in their dogs’ behaviours during veterinary visits and who were interested in making such visits as happy as possible were solicited for the study. Posters were placed in local businesses and veterinary offices within Charlottetown, Prince Edward Island, Canada and within the Atlantic Veterinary College (AVC) Veterinary Teaching Hospital (VTH) waiting room, and in the hallways of the hospital. A recruitment message was sent to AVC staff and veterinary students via the AVC dean’s office. Inclusion criteria specified that participating dogs had to be at least 6 months of age and be in good health. By requiring that dogs were 6 months of age, we guaranteed that all had had some prior veterinary experience. This is also the minimum age for pharmacological studies for behavioural medications, including those used prophylactically for veterinary evaluation. Exclusion criteria included females that were pregnant or lactating, animals that were receiving behaviour-altering medications, and those that had a history of overt aggression during veterinary examinations. Overt aggression was determined via an aggression screen within the pre-enrolment WDQ (Working Dog Questionnaire–pet version) that asked what behaviours their pet demonstrated at the veterinary clinic. Behaviours included snarling, lifting their lip, barking, growling, snapping, biting, withdrawing, or having no reaction.

Owners were told at the end of the study that the laboratory results could be sent upon request to their family veterinary clinic, and if they wished to further pursue improvement in their dog’s behaviour at the veterinary hospital, free visits would be arranged to address their dog’s concerns. Two dogs were removed from participation during the study due to frank aggression (growling, snarling, lunging, snapping) when we attempted to touch them, which prevented their safe handling. This left 28 dogs, the target number, for the control (14 dogs) and intervention (14 dogs) groups.

### 2.2. Questionnaire

Prior to enrolment, pet owners were asked to complete the WDQ—PET questionnaire (Appendix A). This questionnaire has been used in dogs across contexts, and when compared with provocative tests for problematic behaviours (aggression, noise reactivity, fear, and separation anxiety) has been shown to accurately portray patterns of behaviour with a low level of false negatives and false positives [28,29,30]. The questionnaire consists of 78 questions and is broken down into six parts: demographic information, reward/reinforcement-based questions, questions about reacting to the environment, general behavioural patterns, husbandry information, and general behavioural and medical history, which included response to absences and noises, ritualistic behaviour, an aggression screen, and age-related changes. The questionnaire provided scorable historical information—including if the dog was adopted or obtained as a puppy, their training history, plus scorable questionnaires for fear, anxiety and aggression. These data will be used in an analysis of behavioural responses during the veterinary examination as part of another paper in progress. The WDQ-Pet also served as check that participants met the inclusion criteria. 

### 2.3. Study Design

The study took place from July to October of 2021 at the AVC Veterinary Teaching Hospital (VTH). The study consisted of four visits across 8 weeks, with each visit being separated by 2 weeks. Once the dog was scheduled into an appointment slot, each subsequent appointment would occur at the same time of day, in order to remain consistent with diurnal cortisol curves [31]. During each visit, a physical examination was conducted and a blood sample obtained. All dogs were weighed at each visit. All dogs were video-recorded and assessed behaviourally using a Likert scale, similar to those published [15,32,33] (Appendix A), during the following events: (1) when the dog walked into the hospital, (2) when the dog was weighed, (3) as the dog entered the exam room, and (4) throughout the physical examination, during which scoring was carried out at the beginning and end of the exam.

All dogs were randomized into two treatment groups of 14 dogs each, a control and an intervention group, using an online randomization tool (Randomizer.org). Owners were unaware of the treatment assigned until the end of visit four. Both treatments were treated identically at the first visit, when every participant received the control treatment. After the initial visit, different protocols were used for each group when weighing the dog on a scale, during the physical exam and blood draw, for visits two to four, and the type of homework assigned between visits. Table 2 summarizes the major differences between each treatment group. 

#### 2.3.1. Homework Protocols

Clients in each treatment group received homework assignments with their dogs; they were asked to perform these for 5 consecutive minutes, three times a week, during the two weeks between their appointments. Written and video instructions were provided, and clients were asked to complete logbooks noting the time of starting and stopping, and any concerns or observations they had. Clients were asked to bring these logbooks to subsequent appointments. Intervention group members were given a soft blue bathmat and a Lickimat^®^ at the end of their first visit, and they were asked to use these tools to practice parts of a physical exam with their pet. Clients were asked to touch various parts of the dog’s body (neck, abdomen, chest, legs), to encourage dogs to turn their heads to offer their ears to handle, picking up each of their feet, applying non-scented lotion to both the front and back legs where blood draws would typically occur to imitate the application of a lidocaine cream, and briefly applying pressure to leg veins. These steps were both written and demonstrated on a dog in an accompanying video. Control group members were simply asked to pet their dogs in the allotted time, and a video was provided for how to calmly pet your dog (Appendix A).

#### 2.3.2. Scale Protocols

The same walk-on stainless steel scale in the same location was used for both groups and all visits. All dogs were weighed at all visits. For the control group, dogs were weighed on the scale when they first arrived for their appointment. The control group participants were asked to walk onto the bare metal scale that was placed against the wall. For the intervention group, there was a blue yoga mat on the scale, the scale was pulled away from the wall, and treats were used to lure the dog (Figure 1). If the control group dogs would not get onto the scale at any visit, the decision tree in Figure 2 was followed in a step-wise manner, and the level of intervention needed to weigh the dog was noted. Such tiered procedures are scorable for level of intervention required to obtain a weight, and thus indicate where the dog has concerns. Similar decision trees were utilized for the physical examination and blood draw portions (Appendix A).

#### 2.3.3. Physical Exam and Blood Draw Protocols

The same exam room was used for all participants during each visit, and the physical examination and blood draw was performed by the same clinician (CS) for every participant. Both the control and intervention group received the same standardized physical examination (Table 3), which has been used in other studies assessing physical examination [32,33,40]. Owners were present throughout the physical exam and blood draw for both groups. Owners were offered chairs, and water dishes were provided for dogs. In the intervention group, the blue mat was placed in front of the owner’s chair, and they were asked to hold the Lickimat^®^ for their pet. In both groups, owners could give treats and pets as needed, and occasionally assisted with handling (such as holding their pet in their lap if comfortable). 

In the intervention group, the exam started with applying a lidocaine cream (EMLA^®^ 2.5% lidocaine/2.5% prilocaine cream; SOLA Pharmaceuticals, Baton Rouge, LA, USA) on both the cephalic and saphenous vessel regions of the legs. For dogs that had long hair, the hair was parted to apply it to the skin. During the examination, the blue mat was placed in front of the owner and they were instructed to hold the Lickimat^®^ for their pet (Figure 3). The Lickimat^®^ was loaded with the dog’s preferred treats (whipped cream cheese, string cheese, Kong^®^ cheese or liver spay, and/or freeze-dried liver or fish treats) prior to the start of the exam. All dogs in the intervention group, regardless of size, were examined on the floor, or if preferred, in their owner’s lap. For some dogs, the mat was placed on the owner’s lap for traction and comfort. Examination took place when the dog was on the mat. If they left the mat, the examination was readjusted to their preferred position to accommodate the dog’s needs. Following the physical exam, blood was drawn, and the dog was told that they were good and offered a treat. No white coat was worn for any phase of the visit for the dogs in the intervention group. 

In the intervention group, a closed needle butterfly needle system (e.g., Vacutainer^®^ Safety-Lok^TM^ blood collection and infusion set) was used to obtain the blood samples. This allowed for there to be less restraint, as the vein did not need to be held off by another team member (as is required for the open needle system routinely used in veterinary medicine), and the dog could remain in a comfortable standing position (Figure 4). 

In the control group, no lidocaine was applied, no Lickimats^®^ and food treats were used, and no soft, stabilizing blue mat was provided for the dog. Small dogs were examined on the table, and larger dogs were examined wherever clients said was routine for the dog. The clinician wore a white coat for the entirety of the visit for the dogs in the control group. 

The intervention group dogs were weighed after the physical examination and blood draw were concluded, prior to exiting the reception area of the hospital into the parking lot.

All dogs, regardless of group, were offered treats at the end of the appointment while in the room, and while in the parking lot when leaving the hospital, since differential consumption of treats may be informative of the dog’s perception of the experience [41]. 

Because this was a study about reducing distress, a humane care exception was instituted during all appointments. If dogs became sufficiently distressed despite adjustments, further intervention such as applying lidocaine cream for the blood draw (if in the control group; N = 4) or providing anxiolytic support (Sileo^®^-Orion Corporation, Orion Pharma Finland, Espoo, Finland/alprazolam; N = 3 in the intervention group) was offered, with any scoring being restricted to the earlier exam. Owners were always able to withdraw from the study at any point, although none did so.

### 2.4. Physiological Measurements

At every visit, a blood sample was obtained to measure serum cortisol (CORT), neutrophil lymphocyte ratio (NLR), and creatine kinase (CK). At the first visit, a complete serum biochemistry and complete blood count (CBC) was obtained to ensure that the participant was healthy. The last physiological parameter measured was the dog’s heart rate (HR), which was obtained at the time of the physical exam. 

CORT is considered a measure of an acute response to a stressor, reaching a peak within minutes to hours of exposure to the stressor [23]. The cortisol response can also be ongoing or chronic, and is regulated by the HPA axis. The HPA control over CORT is considered to have evolved as an adaptive homeostatic mechanism to allow recovery from stressors [23]. However, not all physiological or behavioural responses are adaptive. When such responses fail, we often see behavioural/psychiatric pathology and maladaptive responses that may signal such pathology. In such circumstances, the outcome of long-term response to stressors may be a blunted/depressed cortisol response [42,43,44,45].

Neutrophil/lymphocyte ratios (NLRs) have also been used as a measure of an individual’s response to a stressor [46,47]. Glucocorticoid (GC) increases, including increases in CORT, cause a rapid increase in neutrophils and a concomitant decrease in lymphocytes, raising the ratio. The change in leukocyte measures has been hypothesized to be adaptive in emergencies, in which combatting infection due to tissue damage is desirable [48]. Because the NLR response to GC stimulation can be delayed, NLRs may not be the most sensitive measure of short-term stressors [49], but NLRs appear to be at least constant over the duration of the stressor [48]. 

In general, the NLR has been found to be a reliable immunological measure of chronic stress, is relatively unaffected by handling associated with acute stress such as blood sampling or potentially confounding factors such a sex or time of day [24], and may be a good marker of sub-clinical inflammation [25]. It was used here as a complement to serum CORT measurement to provide a joint profile of relatively acute stress (the procedure) and some measure of ongoing or more chronic stress (the dog’s perception of life). 

Heart rate responses, specifically tachycardia, have been used in previous studies to measure situational acute stress [50,51,52]. HR as a fear and/or stress response is due directly to activation of arousal as part of the sympathetic response originating in the locus ceruleus (LC). The LC is the primary source of norepinephrinergic (NE) neuronal activation throughout the brain in sympathetic arousal responses, and has inputs to the lateral nucleus, the basolateral nucleus and the central nucleus of the amygdala [53]. The NE produced by the LC acts to trigger somatic physiological responses to the stressor [22], largely through amygdalic stimulation. Stressors of a more ‘psychological’ nature are thought to have a more profound effect on NE release than stressors of a more physical nature such as restraint [54]. Accordingly, HR may be one measure of the individual’s perception of whether a stressor is more ‘psychological’ to the individual. 

CK has been used as a plasma marker of acute muscle damage, including cardiac muscle damage [55], and muscle pain and fatigue due to exertion, acute tension, or illness [56]. In profound panic, acute muscle contraction and injury may be one component of the arousal and stress response associated with freezing [26]. Freezing with muscle contraction and increased CK has been found in dogs undergoing lactate testing for profound anxiety/panic [57].

While HR, CORT and NLR all share overlapping mechanisms initiating their response, CK evaluates a different system and is not a traditional stress response measure. It was included here to ensure that we identified any dogs experiencing panic. Table 4 summarizes the physiological markers and their specific stress measurement. 

### 2.5. Statistical Methodology

A two-way ANOVA for repeated measures was conducted to assess differences between the control and intervention groups for HR, CORT, CK, and NLR. A Wilcoxon Mann–Whitney test was conducted to assess changes in HR, CORT, CK and NLR across the four visits within the control and intervention group. 

There is no single response to any stressor. We chose three potential measures (HR, CORT, NLR) for which the physiological response to stress is well known, and one that measures muscle damage (CK). The effect on the individual is a combined measure of all response patterns. We attempted to assess the combined effect of the stress response indicators experienced by the dogs by creating a dimensionless composite index, the stress response index, that included summed standardized measures for HR, CORT and NLR, using a winsorizing process that removed extreme outliers. Paired t-tests were used to test for differences between the first and fourth visit. 

## 3. Results

### 3.1. Changes between Control and Intervention Groups

There were no significant differences between the control and intervention groups for HR, CORT, CK, and NLR for each visit.

### 3.2. Changes within Control and Intervention Groups

The probability associated with the overall change from the first to fourth visit in cortisol was *p* < 0.08 (based on a Wilcoxon Mann–Whitney test: Z = 1.75). However, when comparing the changes between the first and last visit, the intervention group had a greater reduction in serum cortisol compared to the control group, with a probability of *p* < 0.04 (Wilcoxon Mann–Whitney test: Z = 1.75). (Figure 5 and Figure 6). 

When assessing the other physiological variables (HR, CK, NLR), there was no significant difference between the groups for the change in values from the first to the last visit. Although there was an observed decrease in HR (Wilcoxon Mann–Whitney test: Z = 0.89; *p* < 0.37, two-tail test; *p* < 0.18, one-tail test) and NLR (Wilcoxon Mann–Whitney test: Z = 0.07; *p* < 0.94, two-tail test; *p* < 0.47, two-tail test) from the first to the last visit in both groups, the differences were not statistically significant. 

### 3.3. Stress Response Index

The stress response index was compared between visits one and four. The index did not differ between the first and last visit for the control group, but did differ between the first and last visit for the intervention group (paired t-tests: intervention *t*-value = 2.37; *p* = 0.027; control *t*-value = −1.12; *p* = 0.29) (Figure 7). The effect size is approximately 1, which is considered a large effect (Cohen’s d = (−0.80–−0.08)/0.72 = 0.99. Glass’s delta = (−0.80–−0.08)/0.65 = 1.12. Hedges’ g = (−0.80–−0.08)/0.73 = 0.99). 

## 4. Discussion

The overall goal of this study was to assess whether canine patients that received a veterinary examination designed to minimize stress and fear, and that was collaborative and adaptive to the dogs’ needs, had a greater reduction in distress across four veterinary visits compared to those that received routine handling methods. The targeted low-stress interventions were simple and practical, and were specifically selected so that they could easily be implemented into a busy veterinary practice. Rather than selecting one parameter to assay, such as not wearing a white coat, we altered as much of the entire visit and examination procedure as we could to provide opportunities for adaptive and collaborative care, and pain and anxiety relief. The dog’s experience is the entire visit, not compartmentalized segments. Accordingly, we cannot attribute any of our results to any one intervention, but instead show what happened across four visits over 8 weeks.

No difference was observed between the control and intervention groups for the physiological parameters for each visit. A possible explanation is the large variability within the study population, including baseline anxiety and fear when entering the study. This led to a wide range of individual responses for all physiological values, and a large degree of overlap between each group. It is also important to note that four visits across 8 weeks is a relatively short time in the life of a dog. Regardless, both the intervention and control groups experienced a decrease in serum cortisol when the first and last visits were compared; however, the intervention group had a significantly greater reduction in serum cortisol compared to the control group, by the last visit. This finding suggests that dogs that experienced adaptive, collaborative care and low-stress intervention techniques throughout all four veterinary visits had a greater reduction in stress over time, an effect that may be magnified were such interventions routine. Serum cortisol has been used in numerous studies as a physiological measurement of acute stress in dogs [6,58,59]. It is important to note that a lack of an increase in serum cortisol in a patient does not necessarily mean that acute stress was not experienced. For example, if the animal was experiencing chronic stress, acute stress measures may be blunted [45]. In humans and animals that experience chronic stress, repeated surges of cortisol can result in cortisol dysfunction, which may include depletion of cortisol, insufficient free (unbound) cortisol, impaired cortisol secretion, and/or glucocorticoid receptor resistance [60]. Therefore, it is possible that some of the dogs in the study had blunted cortisol values while still experiencing the same level of stress of those that showed the increased serum cortisol concentration. This phenomenon will be investigated further when combining the behavioural scores and historical data, as it will hopefully help to reveal those individuals whose stress may have been undervalued in this initial data analysis. 

For both groups, there was an increase in serum cortisol in the second visit, followed by a decrease at the third. This may be explained by an initial period of sensitization in the dogs, followed by a period of habituation [61]. The pattern of serum cortisol over time suggests that that both groups experienced a degree of habituation across the four visits, but the effect was greater for the intervention group. The intervention group significantly improved when comparing the last visit to the first visit using the stress response index, but the control group showed a non-significant worsening of response. This result suggests that members in the control group could have been sensitized across the visits, in the absence of any of the calming techniques that the intervention group experienced. 

The stress response index also differed significantly between the first and last visit, but only for the intervention group. The effect size is approximately 1, suggesting that the effect is real, and possibly important, especially since there was a non-significant pattern of the composite stress index worsening at the last visit compared with the first visit in the control group. The non-significant increase in the stress response index for the control group may suggest that when the combined effect of all physiological responses was examined, dogs may have been experiencing some sensitization to repeated visits. Composite scores may be a valuable tool for understanding the global effects of multiple integrated physiological systems. 

The decision to have limited inclusion criteria was important, as the study population aimed to represent the real-life patient population found within veterinary clinics. The study groups successfully mirrored the large variety observed within the AVC setting (and most participants were drawn from AVC patients), including a wide range of breeds, sizes, and histories, in addition to varying baselines of anxiety and fear. A representative population of patients was solicited for participation rather than choosing a group of standard age, size, breed and/or background, to render the outcome more applicable to our patients. Accordingly, our results were highly variable. The size of the dogs alone ranged from a 3 kg Chihuahua to a 64 kg Labrador retriever. The dogs in the study also had a range of veterinary experiences, most of which we could not quantify, which could have contributed to their anxiety and fear. Despite the variability, greater improvement was observed in dogs that received low-stress interventions. 

Because we aimed to not cause extreme duress in the study participants, some degree of an adaptive exam was available to all patients. For one participant, this meant going outside the building for the blood draw so that they would not feel confined in the exam room, and for others, this meant using other intervention methods such as lidocaine cream, even for patients in the control group (N = 4), or anxiolytic medication (alprazolam or Sileo^®^) (N = 3 in the intervention group) [62]. These interventions occurred when the initial treatment for their group, plus routine adjustments (including using treats, adjusting locations, providing breaks, and using other tools such as towels) as outlined in the procedure decision trees, was unsuccessful (Appendix A). The last possible intervention offered was Sileo^®^, an oral dexmedetomidine gel. This medication is an alpha-2 agonist that blunts the release of norepinephrine from the locus ceruleus within the brain and has been used successfully in hospital settings to reduce fear and anxiety in patients [32,33,63]. Sileo^®^ can lower HR, so it is possible that the three participants who received it (all of whom were in the intervention group) may have experienced a very slightly lower HR as a result. In a repeated measures design, each dog acts as its own control, which is inherent in the statistical analysis. These dogs fell well within the variation for other dogs in this group at each assessment (they were not outliers) The dogs in the control group that received lidocaine cream for venepuncture may have experienced beneficial alterations in their physiological values, yet from the first to the last visit, the control group worsened. The effects of the interventions on NLR and CORT are less likely, as the laboratory effects we measured have long half-lives. Behavioural effects, not discussed here, are possible. We wanted to fully disclose this humane care usage.

For both groups, the owners were present throughout the entire appointment, including for the physical exam and blood draw. For the control group, the protocol aimed to match what patients at AVC would typically experience, but having the owner present deviated from the AVC protocol. At AVC, it is standard to remove patients from their owners and take the patient into a hospital treatment room for these procedures. Pets who are able to stay with their owners during examination or other stressful procedures show fewer and lesser signs of distress [10,18,52]. Because we allowed both groups to be with their owners, this likely affected—and lessened—the level of stress the control group members experienced, versus what is experienced by many dogs and cats routinely at veterinary clinics when separated. 

Other limitations included an inability to completely blind the study. Because of the specific protocols and tools required for each group, it was impossible to blind researchers to what group the participants were assigned to. All possible actions were taken to reduce biases. The owners of the dogs did not know which group they were in until the very end, although it is possible that they surmised their group given on the homework and appointment protocols they experienced. Regardless, the laboratory technicians running the blood tests were blinded to the group status of each blood sample.

## 5. Conclusions

The aim of this study was to determine whether interventions affected measurements of distress at the veterinary clinic in canine patients. The initial findings from the physiological data presented suggest that patients receiving low-stress intervention techniques during veterinary visits have a greater reduction in stress over time compared to those without interventions. The use of composite scores such as the stress response index may be a valuable tool for understanding the global effects of multiple integrated physiological systems. Further analyses will investigate the relationship of these findings with behavioural scores and historical data.

## Figures and Tables

**Figure 1 animals-13-01253-f001:**
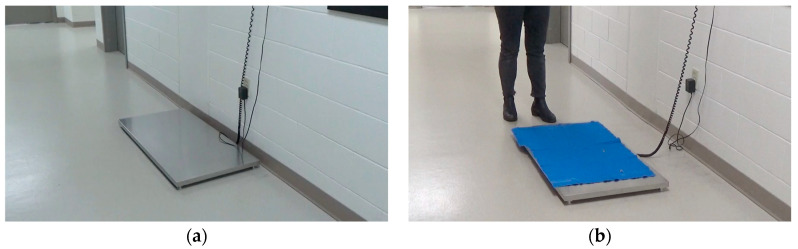
(**a**) Scale set up for control group and (**b**) intervention group. The scale procedure for the control group occurred at the beginning of the appointment; for the intervention group, it occurred at the end of the appointment.

**Figure 2 animals-13-01253-f002:**
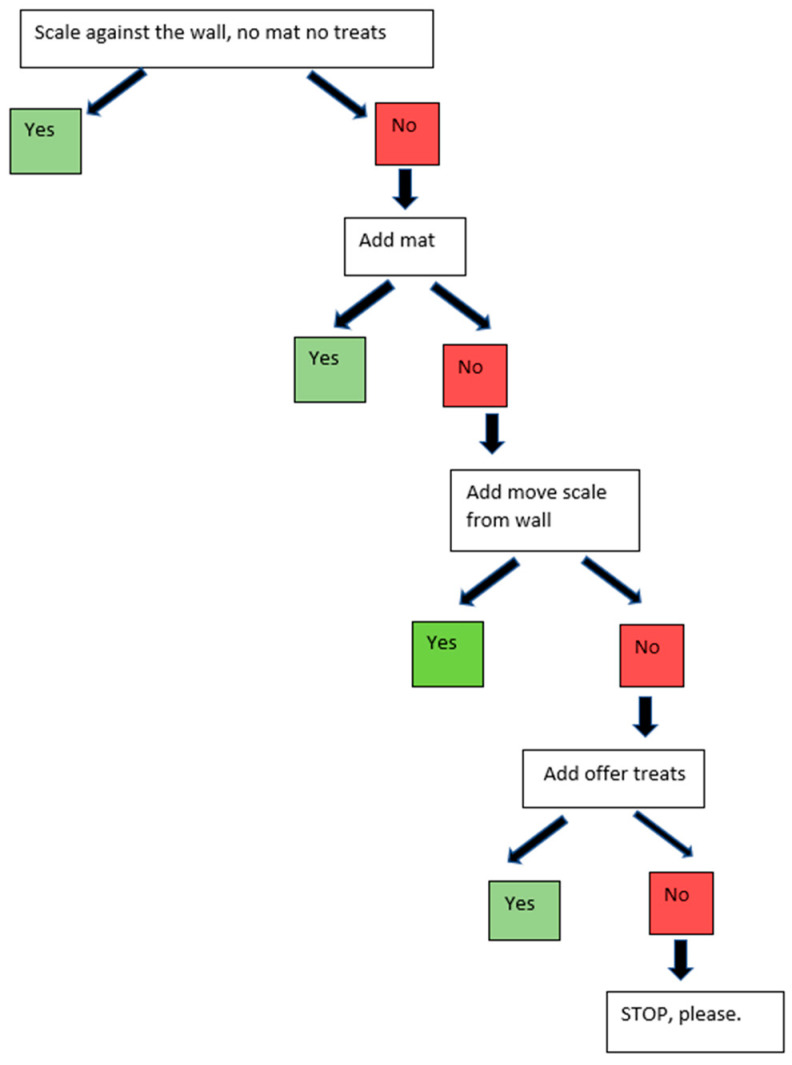
Decision tree for scale procedure for control group (noting the level at which they could be weighed at each visit).

**Figure 3 animals-13-01253-f003:**
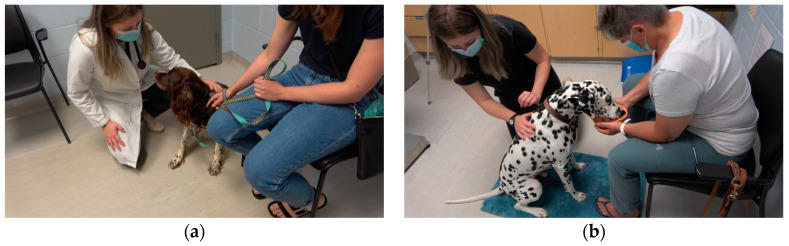
(**a**) Physical exam example for control group and (**b**) intervention group.

**Figure 4 animals-13-01253-f004:**
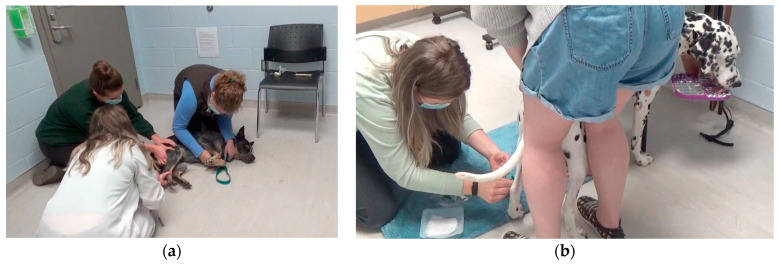
(**a**) Blood collection example for control group and (**b**) intervention group.

**Figure 5 animals-13-01253-f005:**
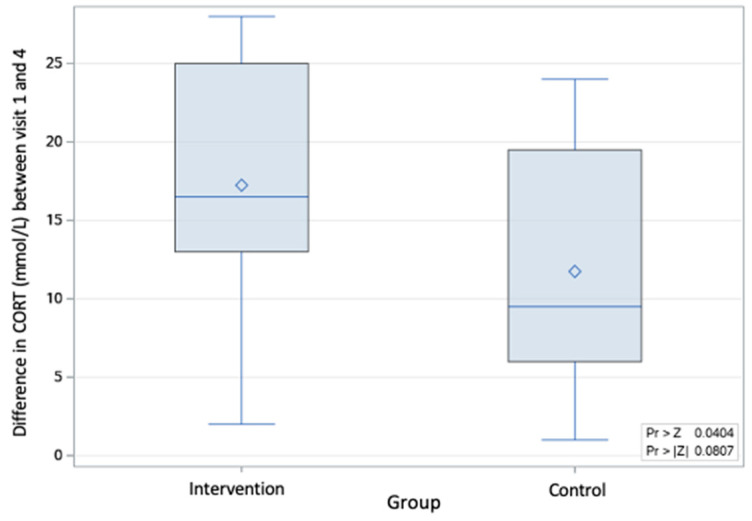
Non-parametric comparison assessing the change in cortisol from the first to fourth visit within the intervention and control groups. The box and whisker plots show the means (rhombus), the medians (lines), and the values for 75% of index scores (whiskers) for each group.

**Figure 6 animals-13-01253-f006:**
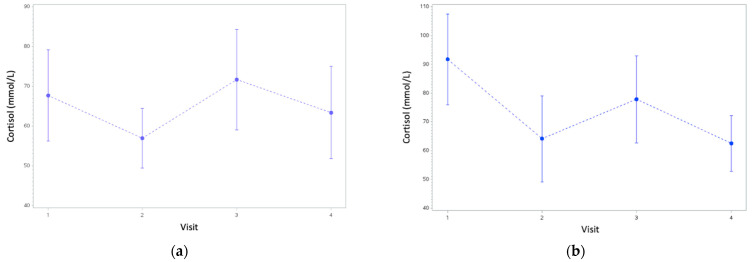
Cortisol values from the first to fourth visit for (**a**) the control group and (**b**) the intervention group.

**Figure 7 animals-13-01253-f007:**
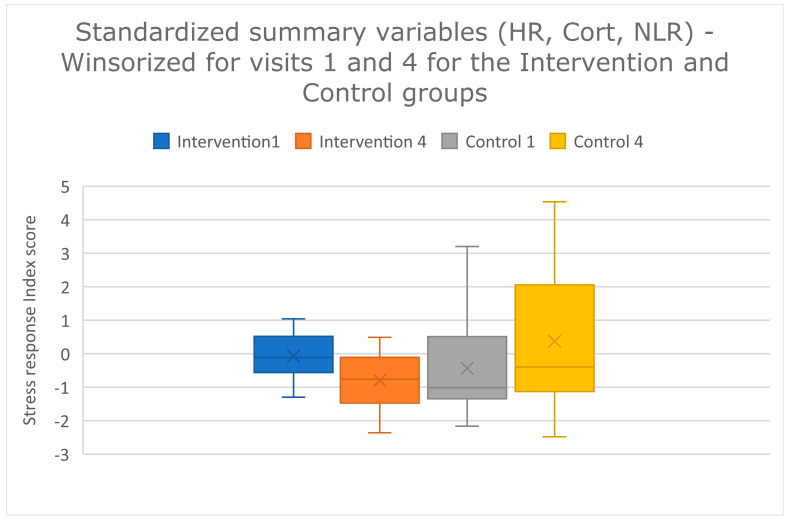
Stress response index score; Summed, standardized measures for HR, CORT and NLR for the first and last visit within the intervention and control groups, as noted in the key on the graph, after data were winsorized. The index score is on the Y axis. Groups are colour-coded by treatment (intervention v. control and visit number (1 v. 4). The box and whisker plots show the means (×), the medians (lines), and the values for 75% of index scores (whiskers) for each group.

**Table 1 animals-13-01253-t001:** Subject signalment, group, and study completion status (MC = male, castrated; MI = male, intact; FS = female, spayed; FI = female, intact).

ID	Age in Months	Sex	Breed	Weight	Group	Status
01	76 mo	MC	Dachshund	7.2 kg	Control	Completed
02	35 mo	FS	Chihuahua mix	5.4 kg	Control	Withdrawn
03	102 mo	MC	German Shepherd/Husky mix	30.2 kg	Control	Completed
04	46 mo	MC	Springer Spaniel mix	19.2 kg	Intervention	Completed
05	61 mo	FS	Beagle	13.2 kg	Control	Completed
06	73 mo	FS	Shih Tzu mix	8.2 kg	Control	Completed
07	31 mo	MC	Pitbull mix	30.4 kg	Intervention	Completed
08	120 mo	MC	Dalmatian	27.0 kg	Intervention	Completed
09	31 mo	MC	Bernese Mountain Dog	38.8 kg	Intervention	Completed
10	15 mo	MC	Maltese/Lhasa Apso mix	4.6 kg	Intervention	Completed
11	38 mo	FS	Springer Spaniel	21.2 kg	Control	Completed
12	48 mo	MC	Dalmatian	29.4 kg	Intervention	Completed
13	84 mo	FS	American Staffordshire Terrier Mix	20.8 kg	Control	Completed
14	24 mo	FS	Belgian Groenendael Sheepdog	21.2 kg	Control	Completed
15	31 mo	MC	Mastiff mix	34.0 kg	Intervention	Completed
16	50 mo	FS	Chihuahua mix	3.0 kg	Intervention	Completed
17	80 mo	FS	German Shepherd mix	27.6 kg	Control	Completed
18	29 mo	FS	Labradoodle	28.8 kg	Control	Completed
19	160 mo	MC	Golden Retriever	38.8 kg	Intervention	Completed
20	68 mo	MC	Poodle Dachshund mix	11.4 kg	Intervention	Completed
21	102 mo	MC	Labrador Retriever	64.0 kg	Control	Completed
22	127 mo	FS	Chihuahua mix	4.2 kg	Control	Completed
23	7 mo	FI	Golden Retriever	23.6 kg	Control	Completed
24	42 mo	MC	Labrador mix	20.6 kg	Intervention	Completed
25	19 mo	FS	Terrier mix	6.6 kg	Intervention	Completed
26	114 mo	FS	Newfoundland dog	56.2 kg	Control	Completed
27	62 mo	FS	Labrador Retriever	25.0 kg	Intervention	Completed
28	24 mo	FS	English Bulldog	25.8 kg	Control	Withdrawn
29	52 mo	MC	Toy Goldendoodle	6.5 kg	Control	Completed
30	9 mo	MI	Australian Cattle Dog	21.5 kg	Control	Completed

**Table 2 animals-13-01253-t002:** Study protocol summaries for control and intervention treatment.

	Control Treatment	Intervention Treatment
Scale (See Figure 1)	Walk-on stainless steel scale Weighed before examination	Walk-on stainless steel scale covered with blue, non-slip yoga mat, moved away from wall, with dog lured on with treats [15,16] Weight after examination [15]
Physical exam	White coat Small dogs on table	No white coat [34,35] All dogs on the floor or client’s lap if that was the dog’s preference [1] Lickimat^®^ (Innovative Pet Products PTY, Australia) and blue mat for non-slip examination [15,16,17,36]
Blood draw	Fake lidocaine application to 3 legs Standard needle and syringe Standard restraint	Application of lidocaine cream to 3 legs [37,38] Closed double ended butterfly catheter system [39] Reduced to no restraint primarily using guidance and positioning [15,16,17]
Homework	Petting dog	Practice the steps of a collaborative physical exam [15,16,17,20]

**Table 3 animals-13-01253-t003:** Exam structure—including order and timing or frequency—used for both intervention and control groups [32].

Physical Examination Protocol
1. Dog stroked gently from head to base of tail three times
2. Hand placed over the thigh pulse point for 30 s
3. Lidocaine (2.5% lidocaine/2.5% prilocaine) put on legs (two saphenous and one cephalic) for intervention dogs (control dogs are just touched in these areas)
4. Auscultation of heart and lungs 15 s from each side of the chest
5. Manual manipulation of lymph nodes (in order submandibular, prescapular, popliteal)
6. Gentle abdominal palpation undertaken for 15 s
7. Each paw lifted for 5 s for testing placement; first hind limbs and then fore limbs
8. Lifting of upper lips (control of the oral mucous membranes)
9. Observation of external ear canals for 5 s each (without an otoscope)
10. Ear thermometer placed in position until reading
11. Eyes examined directly (observation of the conjunctiva, checking of the cornea) for 5 s each
12. Venepuncture
13. Gently put your hand on the dog’s back and tell them they are good
14. Remove from table and give treat, or if the dog is on the floor, just give the treat (note whether the dog takes the treat on record)
15. Walk client to parking lot and give treat mid-way to car (note response on record)

**Table 4 animals-13-01253-t004:** List of selected physiological markers and what specific component of a stress response each evaluates.

Physiological Marker	Stress Measurement
Serum cortisol	Acute stress
Neutrophil lymphocyte ratio	Chronic stress associated with inflammation
Heart rate	Acute stress—immediate sympathetic response
Creatine kinase	Muscle damage associated with panic response

## Data Availability

The data presented in this study are available on request from the corresponding author. The data are not publicly available due to the ethical agreement with the participants that only anonymized, summary data would be publicly presented.

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
