# Peer review of "Effects of Changing Veterinary Handling Techniques on Canine Behaviour and Physiology Part 1: Physiological Measurements"

_animals, 2023, doi:10.3390/ani13071253_

Round 1
Reviewer 1 Report
This paper is very well done, and it describes a new protocol for reducing stress in dogs at the vet clinic. I really enjoyed reading this and I think it makes a useful contribution to the literature. I do have a few suggestions for the authors.
Title – the authors say ‘canine behaviour and physiology’, but they didn’t measure behaviour, only physiology. This should be changed.
Keywords – as above: ‘canine behaviour’ should be removed because it wasn’t measured.
Intro
L122- what do PEI and AVC stand for? Where is Charlottetown?
L126 – any reason why the dogs had to be at least 6 months old? Please justify this because dogs will have already (hopefully) had several veterinary experiences by 6 months old, for vaccinations etc, so they may already have had negative experiences that impact the baseline measures. It’s not a dealbreakers because there were repeated measures, but a justification of the age limit would be helpful.
Methods
Table 2 - it looks like most of the intervention treatment elements were very simple swaps that would be easy to integrate into any veterinary clinic, and wouldn’t take any, or very much, more time than a standard consult as described in the control treatment. Most of the onus is on the clients, but even that’s not too intensive. I think this is a major strength of this approach, so I would recommend that the authors highlight this in the simple summary and abstract, and even in the title, if possible. My initial concern before reading the paper was that the intervention would be unrealistic to implement for typical vet clinics, but that fortunately does not appear to be the case. Make that a central theme of the entire piece because you don’t want veterinarians to discount the research out of hand and not even read it, just because they assume it won’t be feasible to implement.
L251-256 – please advise how many dogs of which groups received each of these additional supports. This is provided in the Discussion but should be made available here.
L264 suggest changing ‘the assault’ to ‘exposure to the stressor’ or ‘the stressful event’
L296 – ‘CK has been used AS a plasma marker…’ The ‘as’ is missing.\
L303 – ‘differ’ should be ‘different’
Table 4 – suggest modifying to say ‘muscle damage consistent with panic response’, or similar, so the reader knows that the primary interest is in panic, rather than just muscle damage with other causes. The tables and figures should operate as stand-alone items, so it shouldn’t be necessary for the reader to get crucial information from the text.
Results
L331 – p < 0.08 is not significant, but the way it’s written implies that it’s been interpreted as significant. What was the alpha level applied? If it was 0.05, then the result is non-sig. If it’s higher than 0.05, this needs to be explained, and justified, in the methods section.
L349 – good job adding the effect sizes. This is useful information that is often omitted in research reports. However, I recommend adding a brief interpreting statement, explaining whether the reported effect size is small, medium, or large.
Discussion
L364 – suggest adding in here somewhere that this suite of alterations was simple to implement.
L389 – ‘during across’ one of these should be deleted.
Reviewer 2 Report
Broad comments
A very well-written manuscript and valuable research! Experiment was clearly discussed, and literature was well referenced. The manuscript could improve with providing additional clarifications and justifications in some sections of their methodology (see below for specific comments). Overall, this paper deserves to be published as it provides further evidence that dog welfare associated with veterinary care needs to be improved.
Specific comments
Simple summary and Abstract
For numbers, revise to be consistent with how they are presented (typical rule of thumb is numbers from one to nine should generally be spelled out, and 10 or above written as numerals.
Introduction
A recent study evaluated the influence of an intervention (owners had to conduct cooperative care on their animals for 4 weeks) on dog veterinary fear levels (stellato et al., 2019); it would serve the paper to incorporate this into the introduction.
Methods
How was your physical examination and procedure developed? Was it based off of what is currently done? On the teaching hospital? Other similar studies?
L120-121: was this part of the inclusion criteria or was this just a method of recruiting dog owners to get them interested in participating? If not part of the inclusion criteria, did you assess dog fear during the first exam to make sure that you didn’t just have all fearful dogs? The aim of this paper is worded as ‘to prevent’ and not ‘reduce’ fear, so I believe this distinction should be made and at least justify or explain any bias in the sample that could have arisen with this method of advertisement.
L128: how was overt aggression determined? Assuming you gave a recruitment survey and they had to report if their dog displayed aggression? Please make this clear and also specify what behavioral definitions were provided for the owners.
L133: suggest correcting to “if they wished”
L136: clarify what is meant by ‘frank aggression’.
L139: Suggest authors spell out WDQ-PET for the readers
L149: should be ‘baseline scores for fear and anxiety’.
Also, how were these baseline scores measured? Was it based on a scale you provided to the participants? Based on subjective owner reports? This is especially important to clarify if it will be used to compare against their behavioral response during the examination.
Why were these specific questions asked (excluding the inclusion criteria questions) if not incorporated in analysis? It would be interesting to see through some modelling if responses to some of these questions (e.g., training history, baseline scores) are associated with their dog’s stress levels during the exam. Is this something that will be analyzed in a follow-up study, then this should be clarified.
L160: how was this likert scale developed? Were there peer-reviewed resources used to guide this development?
Table 2 –it would be best to insert justification for each intervention treatment in the section above the table presentation and cite relevant articles where appropriate (e.g., based on commonly recommended practices etc.)
L208: please report who performed the examination / blood draw procedures
L212: where were owners located in the room? Did they intervene or assist in any way?
Table 3, L13: revise to “tell them they are good”
L322: The r software and version should be moved to when the analytical software is first mentioned (L113).
Results
Figure 6: Suggest this be made into bar graphs so that the data is easily presented to readers.
Figure 7: Suggest removing the word ‘sum’ from all legend colors to simplify the presentation.
Discussion
Authors should provide some hypotheses as to why they found no difference between physiological parameters between control and intervention groups for each visit. Do we think this is a result of individual differences within each visit or a persistent level of fear found between both groups despite efforts applied to alleviate stress?
L361: As mentioned above, please clarify if the aim was to prevent or reduce? These are two different tasks.
Reviewer 3 Report
I applaud the authors for working to improve the health and wellbeing of dogs visiting veterinary clinics, through techniques that are humane and effective. The paper is well-written, with excellent grammar, and it is easy to comprehend. However, I have some serious concerns about the study design that were not addressed.
Introduction
The introduction was fairly complete. However, there glaring omission of references to other methods of alleviating anxiety in pets visiting veterinary clinics, including pharmacological. It is especially concerning when you gave some of the dogs pharmacological intervention, with no references to these medications in the introduction. While not the focus of this study, the lack of mention of these studies is apparent. Other references to consider are below, plus the ones on dexmedetomidine.
https://www.mdpi.com/2076-2615/12/2/187
https://journals.plos.org/plosone/article?id=10.1371/journal.pone.0215416
https://avmajournals.avma.org/view/journals/javma/251/2/javma.251.2.195.xml
https://avmajournals.avma.org/view/journals/javma/260/9/javma.21.03.0167.xml
https://avmajournals.avma.org/view/journals/javma/260/8/javma.20.10.0547.xml
https://www.sciencedirect.com/science/article/pii/S1558787820300447?casa_token=BJcQQgRYqu0AAAAA:5ere8AIDZ9j0sAxHicbCygfn03W7eLNWvF2iplzfbfXt89pCd7-nxDYAAXt9TSISTQEH-oJ5UOM
https://www.ncbi.nlm.nih.gov/pmc/articles/PMC8360309/ (a review of preappointment medications; however, it doesn’t address the most recent studies)
It is also important to state that this is now standard of care, per the AAHA guidelines: https://meridian.allenpress.com/jaaha/article-abstract/51/4/205/183314/2015-AAHA-Canine-and-Feline-Behavior-Management?redirectedFrom=fulltext
Methods
You laid out the study design very clearly and it was easy to understand.
You stated that the WDQ_PET was validated. “Validated” is a highly inappropriate word for this, as none of the three studies referenced actually validated the survey, and one of the authors was an author on all three referenced studies.
Study design: While I applaud you looking at all of the interventions, there is no way to determine which of these variables had the positive effect. There were way too many to evaluate. Just one example: the scale protocol was different in two major ways: the use of a mat; and the timing of weighing.
I am curious as to why you didn’t use HRV in this study. While you do have references for changes in HR related to stressful events, it should not be considered a reliable method. Additionally, while you used CORT and referenced appropriately, there are references that also show that this should not be considered an infallible method of measurement of stress.
I am also curious as to why you didn’t evaluate the videos of the behavioral assessments, as outlined in in Line 159+ and S2. You mention doing them and then there is no further mention of them. This is another error of omission.
Line 254+: You mention Sileo and alprazolam, yet still there are no references. Also, I believe that you should not be using the brand name in the paper; instead, it should be transmucosal dexmedetomidine.
Regarding medical intervention, were these dogs given pharmacological help during both interventional visits? I would suggest evaluating those dogs separately, because they could have accounted for the lessening of the stress response between visit 1 and visit 4.
There was no evaluation of order effect. There is no explanation of why this study wasn’t a crossover trial. It is also inappropriate to use a one-tailed test, seemingly at random.
There is no discussion of a power analysis.
Results
I am seriously concerned about the flip-flop between using a 2-tailed, then a 1-tailed, test in paragraphs 328 and 339. This is highly inappropriate. You also state that, while there was a change in CORT over time, you didn’t state that it wasn’t significant (p < 0.08) (line 331). Figures 5 and 6 do not reflect the reality of the results.
I am confused as to why, in some places, you state p values as p < xxx, and others you state them as p=xxx, when you are discussing nonsignificant results.
You start discussing the ‘stress response index’ in the results on Line 346 but never discuss what it is. Is it HR, CORT, NLR? You call it “indicators” in Line 318.
Discussion
You approached some of the shortfalls of the study in the discussion. However, there were glaring omissions.
You glancingly addressed the many different interventions in the study (first paragraph in discussion). You do not address the fact that there were no differences between groups, only that the interventional group had changes between visit 1 and visit 4.
In the discussion, you should address the owners’ beliefs around handling techniques. You can again reference the Stellato papers, along with https://www.mdpi.com/2076-2615/12/11/1387. Another point of discussion is how to address veterinarians’ willingness and ability to perform all of these techniques. What if giving medication is just as effective in reducing stress, and it’s easier for the owner? Of course it isn’t the only answer, and lower-stress handling is best, but I am concerned that veterinarians will read this and see that they have to do ALL of these techniques, and essentially doing none. One reference to cite for veterinarians’ perception of behavior in practice is https://bvajournals.onlinelibrary.wiley.com/doi/full/10.1136/vr.101124
Line 404: you state that this population mirrors the population seen in veterinary clinics, yet there is no reference.
Line 439: Add the Roshier, et al paper here.
Round 2
Reviewer 3 Report
Thank you for the responses and changes.
Point 1: I understand the reason for not addressing pharmaceutical interventions in the introduction, but the authors could have summarized them in 1 sentence with references, along the lines of "In addition to different handling methods, pharmacological interventions have also been demonstrated to help alleviate stress and anxiety." (refs x-z)
Point 4: I understand that HRV is not something a veterinarian would use in a standard setting, and I am fine with it not being utilized in this study. However, the justification that it isn't used by veterinarians in practice is weak. This is research and HRV would have been another measurement of changes in stress. It should be addressed in the discussion as another method that could have been used to measure physiological differences.
Point 6: Thank you for adding the references for Sileo and alprazolam. Other medications, and their references that are from authors aside from the last author, should be mentioned for completeness.
Point 8 and 10: It appears now that you removed the verbiage for 1-tailed test from the paper (delete line 367). While the expectation is that dogs would have improved scores on the second visit, you cannot assume this. All tests should be a 2-tailed test.
